## Research Article

coastal flooding; coastal geomorphology; coastal morphodynamics; coastal risk; numerical modelling

**Corresponding author:**
Gerd Masselink;
Email: g.masselink@plymouth.ac.uk

# Numerical modelling of the 1 July 2022 flooding event, Southwest Huvadhoo Atoll, Maldives: Implications for the future

Gerd Masselink[1] 🔘, Tim Poate[1], Timothy Scott[1], Floortje Roelvink[1,2] and Robert McCall[2]

[1]Coastal Processes Research Group, University of Plymouth, Plymouth, UK and [2]Deltares, Delft, Netherlands

## Abstract

Low-lying atoll islands are among the world's most vulnerable coastal environments to sea-level rise (SLR). Global application of coastal flooding models suggests that centennial flood events may become annual events by 2050 in tropical regions. This article addresses this claim by modelling an island flooding event that occurred in the Maldives on 1 July 2022 as a result of a distant-swell event coinciding with an extra high spring tide. Hydrodynamic data collected after the event on one of the affected islands were used to calibrate and validate a one-dimensional non-hydrostatic XBeach model. The model overpredicted wave setup and underpredicted the water motion at frequencies <0.05 Hz, but the wave run-up elevation was predicted reasonably well. The 1 July flood event was considered in a decadal context using modelled wave data and measured tide data. It was concluded that the 1 July event represents a c. 1:25-year flooding event, but, due to SLR, such flooding could occur every few years by 2050. This prediction ignores natural or anthropogenic adjustments to the island morphology. The expected increase in frequency of coastal flooding in the Maldives requires atoll and island authorities in the Maldives to act swiftly in adapting to future flood risk.

## Impact statement

Low-lying atoll islands are among the world's most vulnerable places on Earth due to rising sea levels. Global computer modelling studies suggest that by 2050, what used to be rare, extreme floods could happen every year in tropical regions. This study examines those predictions by analysing an island flooding event in the Maldives that occurred on 1 July 2022. The flooding was caused by long-period swell waves coinciding with an extremely high tide level and affected 20 islands. The flooding event was simulated with a computer model, and the results were compared with field observations. The model did not get everything right, but it was reasonably accurate in predicting the height reached by the waves and the occurrence of flooding. By looking at all the major storms that occurred over the period 1990–2023, it was concluded that the 1 July flooding event was a relatively rare event. However, with sea levels rising, such flooding could occur every few years by 2050. This prediction ignores any natural or anthropogenic adjustments to the island morphology. Although this study was conducted for one island only, the results have application to other Maldivian islands. This study implies that the expected increase in flood frequency and magnitude over the next few decades requires authorities in the Maldives to act swiftly in assessing future flood risk. Based on such assessments, viable adaptation strategies to mitigate against the adverse impact of flooding need to be identified, evaluated and implemented without delay.

## Key points

- A distant swell event coinciding with spring high tide resulted in flooding of 20 islands in the Maldives on 1 July 2022.
- The event was simulated using one-dimensional XBeach-NH calibrated and validated with a hydrodynamic data set collected on one of the flooded islands.
- The 1 July event can be considered a 1:25-year flooding event, but due to sea-level rise, such flooding could occur every few years by 2050.

## Introduction

Atoll islands are wave-built accumulations of gravel or sand that sit on top of coral reef platforms. The existence of these islands is intrinsically linked to the reef ecology, as they rely on the reef's production of sediments; however, their formation, maintenance and dynamics are primarily

governed by physical processes involving sea level, waves and currents (Kench, 2014). Key hydrodynamic processes include: (1) shoaling and frictional dissipation of waves across the forereef (Lowe et al., 2005; Monismith et al., 2015); (2) loss in wave energy as waves break on the forereef and in the shallow water across the reef platform (Brander et al., 2004; Lowe et al., 2005; Monismith et al., 2013; Beetham et al., 2016); (3) generation of long-period waves and wave set-up across the reef (Pomeroy et al., 2012; Cheriton et al., 2016; Masselink et al., 2019; Cheriton et al., 2020; Cheriton et al., 2024); and (4) the combined wave motion results at the island beach in wave run-up (Shope et al., 2017; Quataert et al., 2020; Scott et al., 2020). If the waves are energetic and the water level is high, the run-up may extend to the top of the island, resulting in (5) overwash and flooding of the island (Beetham & Kench, 2018; Hoeke et al., 2021).

A characteristic feature of atoll islands is their low-lying nature (<2–4 m above mean sea level), which makes them vulnerable to coastal flooding and island inundation during episodic extreme wave and water-level events. Sea-level rise (SLR) is expected to significantly increase the frequency and intensity of coastal flooding, shoreline erosion and saltwater intrusion into the freshwater lens of atoll islands. The most recent report by the Intergovernmental Panel on Climate Change (IPCC, 2021) predicts a global mean SLR by 2100 (relative to 1986–2005) of 0.44–0.76 m according to the intermediate emission scenario SSP2-4.5. Several recent studies have used such sea-level projections together with wave transformation models to forecast future flooding on specific atoll islands (Beetham et al., 2016; Storlazzi et al., 2018; Brown et al., 2020; Amores et al., 2022) or atoll islands in general (Quataert et al., 2015; Beetham et al., 2017; Pearson et al., 2017; Beetham & Kench, 2018). Global application of flooding models further predicts that extreme sea levels (i.e., those that have a 1:100 probability of occurrence) will become increasingly common due to SLR (Taherkhani et al., 2020). It is suggested that, even under a modest SLR scenario, a large part of the tropics, where all atoll islands are located, would be exposed annually to the present-day 100-year event from 2050 (Vousdoukas et al., 2018).

Hydrodynamic models used for predicting the impact of SLR on atoll islands invariably assume that the islands are inert features and that there is no morphological adjustment to the island. This is an oversimplification, as many observational studies have reported on the physical and morphological impacts of extreme wave and water-level events leading to atoll island inundation and overwash, either as a result of tropical cyclones (Maragos et al., 1973; Baines & McLean, 1976; Scoffin, 1993; Kayanne et al., 2016; Duvat & Pillet, 2017), distant swell events (Hoeke et al., 2013; Smithers & Hoeke, 2014; Wadey et al., 2017; Duvat et al., 2020) or tsunami (Kench et al., 2006; Gischler & Kikinger, 2007; Kan et al., 2007; Kench et al., 2008; Hoeke et al., 2013). These morphological impacts have also started to become modelled physically (Tuck et al., 2019a; Tuck et al., 2019b; Tuck et al., 2021) and numerically (Masselink et al., 2020; Roelvink et al, 2025; Yao et al., 2025), and can increase island resilience. Ignoring the morphological impacts of SLR on atoll islands, it is widely considered that many of the atoll islands will be uninhabitable by the mid-21st century because of SLR exacerbating wave-driven flooding (Storlazzi et al., 2018).

The general aim of this study is to present field observations and numerical model results of atoll island flooding as a result of a distant-swell event coinciding with an extra high spring tide using a locally calibrated and validated numerical model, and apply this model to investigate future flood risk due to SLR. The flooding event, which occurred on 1 July 2022 (referred to as the '1 July event'), affected more than 20 islands across several of the southern atolls of the Maldives and was the most significant flooding event in the region since at least 2007. Other studies have investigated increased island flooding in the Maldives due to SLR (Amores et al., 2021; 2022); however, these previous investigations have used uncalibrated hydrodynamic models and idealised topographic profiles. The specific objectives of this article are to extend previous studies on flooding in the Maldives by: (1) documenting the relatively rare 1 July event and probabilistically characterising the oceanic forcing conditions (Section title "*Study area and 1 July event*"); (2) using data collected from one of the flooded islands just after the flooding event to describe the hydrodynamic conditions across the reef platform (Section title "*Results field campaign*"); (3) reproducing the flooding event using a phase-resolving numerical model (XBeach-NH) that is calibrated and validated with the local hydrodynamic data (Section title "*Results of numerical modelling*"); and (4) applying the numerical model to all 158 storms that occurred over the period 1993–2022 and evaluating island flooding risk in the past, present and future (Section title "*Application of the model to past, present and future island flooding*").

## Study area and 1 July event

### Description of study area: Geography and oceanography

Huvadhoo Atoll is the southernmost of the large atolls in the Maldives (Figure 1a, b), and the main study site, the island of Fiyoaree, is located along the most exposed southwest (SW) rim of the atoll. Fiyoaree has much in common with the inhabited islands of Rathafandhoo to the west and Fares-Maathoda (Figure 1c) to the east, which are also considered in this study. Fioyaree is 1,600 m long and 600 wide, and is fronted by a c. 200-m wide reef platform. The island has a population of c. 1,500. Outside the build-up area (Figure 2c), the island consists of a mixture of farmland (southeast [SE] and northwest [NW] of the village; Figure 2b), palm trees (NW tip of the island) and mangroves (SE tip of the island). Most of the island is elevated 1–1.5 m above mean sea level (MSL), but a part-natural and part-maintained gravel ridge is present around the margin of most of the island, especially along the most exposed part (Figure 2d,e). As part of a groundwater survey of Fiyoaree commissioned by the Maldives Government, the elevation of the island crest was surveyed at various locations along the ocean beach and indicated a typical elevation of 1–1.5 m MSL for the NW and SE margins of the island, and c. 2 m MSL for the central part of the island (MoEnv., 2020). The ocean sandy beach is widest along the central part of the island (Figure 2d), and is replaced by a coral rubble beach towards the NW and SE tips of the island. A sandy beach is also present along the lagoon shore (Figure 2a).

The reef platform in front of the island has an elevation of c. -0.5 m MSL and is covered by water under practically all wave and tide conditions. The platform is relatively devoid of sediment; gravel sheets/ridges, found on some of the neighbouring islands and in the shallow passages between islands, are not present. The platform is either bare rock, or covered with turf algae or seagrass (Figure 2f). The only significant sediment store on the platform is a c. 0.2-m high and 10-m wide seagrass-covered 'bar' or 'terrace' present just seaward along the toe of most of the beach, and mainly composed of silt and fine-sand material. Live coral is present on the reef platform mainly within localised depressions in the platform that are permanently submerged. At the seaward margin of the reef platform, an extensive and elevated region with crustose coralline algae and other corals is present. Seaward of this region, the

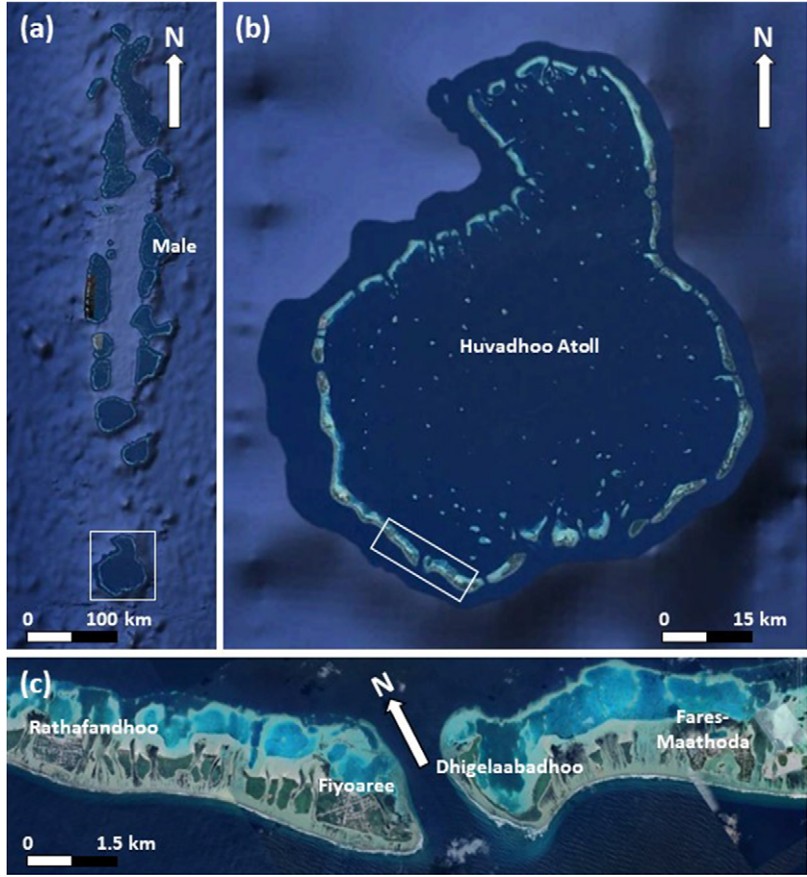

**Figure 1.** (a) Map of the Maldives with box indicating Huvadhoo Atoll. (b) Satellite image of Huvadhoo Atoll with box demarcating studied atoll islands. (c) Satellite image of Rathafandhoo, Fiyoaree, Dhigelaabadhoo and Fares-Maathoda. Source: GoogleMap.

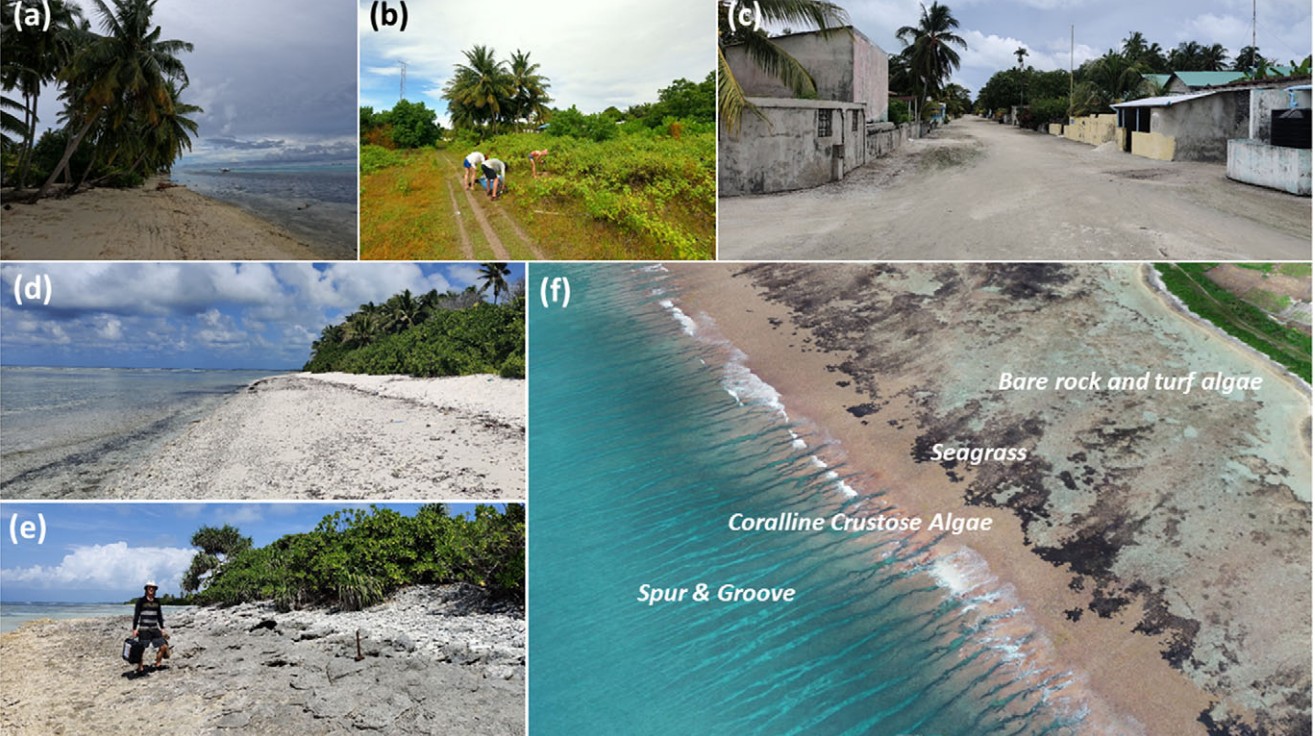

**Figure 2.** Representative photos taken from key environments on Fiyoaree. (a) Lagoon beach; (b) agricultural area in the island interior; (c) centre of the village; (d) sandy ocean beach; (e) conglomerate platform with coral rubble ridge along the ocean-side flanks of the island; and (f) drone photograph of reef platform fronted by forereef characterised by an extensive spur-and-groove system. Source: Authors.

topography drops down steeply with a gradient of 0.08 across the heavily spur-and-grooved and highly rugose forereef (Figure 2f). The spur-and-groove topography is highly variable, but a typical spacing is 10–15 m, and the maximum depth of the grooves relative to the spurs can be up to several meters.

The Maldives are micro-tidal, and the mean neap and spring tide range at the study site is 0.5 and 1 m, respectively (based on tide gauge record from Gan, located 100 km south of the study area on Addu Atoll, but considered representative of the tides in the study area). Although the region can be affected by distant cyclone-generated waves, it is located outside the cyclonic belt, and surge levels are generally <0.15 m (also based on Gan tide gauge data). The Maldives are affected by four types of wave conditions (Amores et al., 2022): SW Monsoon, SW Swell, SE Swell and NE Monsoon. The largest waves affecting the SW rim of Huvadhoo Atoll (including Fiyoaree) are SW Swell, and these tend to be maximised during the period June–August, with typical and maximum wave heights of 1 and 2.5 m, respectively. According to a modelling study by Amores et al. (2021), the 1 in 100-year $H_s$ associated with SW swell at the study site is c. 3.5 m.

Despite the low elevation of Fiyoaree (generally <2 m above MSL), island flooding (by ocean waves) is a rare occurrence. In recent times, significant island flooding occurred in September 1987, December 2004 (Boxing Day tsunami) and May 2007, and, throughout the Maldives, a combination of high tides with large waves and long periods (>15 s) is considered necessary to induce flooding (Wadey et al., 2017). Under energetic wave conditions coinciding with high tide, the gravel ridge along the island's margin does get activated regularly, but its elevation and dense vegetation generally prevent significant island flooding.

### Climatic and oceanographic conditions during the 1 July event

The 1 July flooding event was the result of a combination of high water levels and energetic, long-period waves, but measured water level and wave conditions for the study site are not available. Therefore, water level data were obtained from the Gan tide gauge, and wave data were obtained for a grid cell SW of Fioyaree [73°, 0°] from the Copernicus Marine Environment Monitoring Service (CMEMS) Global Ocean Waves Analysis and Forecast [Global Ocean Waves Analysis and Forecast|Copernicus Marine Service (https://data.marine.copernicus.eu/product/GLOBAL_ANA LYSISFORECAST_WAV_001_027/description)], representing a water depth of 1,000 m, based on the Météo-France WAve Model (MFWAM) (Ardhuin et al., 2010).

The sea level during the 1 July event was placed in a wider context using the full sea-level data set (1990–2023) from Gan (Figure 3). Sea level at Gan is considered representative of the water level at the study site (with a 30-min time lag), based on a comparison between the Gan sea-level data and that measured in the harbour of Fares-Maathoda located 6 km east of Fioyaree over several months in 2022 (not shown). The SLR over the 1990–2023 period of c. 15 cm recorded by the Gan tide gauge (Figure 3a) represents a rate of SLR of just over 4 mm/yr. The 1 July event was characterised by a maximum water level of 0.53 m MSL. This represents an exceedance probability of 2.5% when considering all high tides of 2022 (Figure 3e). When the complete 34-year sea-level time series of Gan is considered, the 0.53 m MSL represents an exceedance probability of 1.2%. The 2-, 10- and 100-year return period water levels for Gan, computed using the Generalised Extreme Value (GVE) distribution, are 0.57, 0.63 and 0.66 m MSL, respectively. The tidal residual for Gan during several

weeks around the 1 July event was 0.15 m (Figure 3d). Although this is relatively modest, it is the highest non-tidal residual that occurred during 2022 (Figure 3c). Inspection of satellite data indicated that the non-tidal residual was associated with a large 'bulge' of warm water to the SW of Maldives that moved to the southern atolls of the Maldives (not shown), causing unusually high water levels for more than a week. The residual tide at Gan for the period March–July (coinciding with the SW monsoon) is generally associated with a positive residual, with a mean monthly value of c. 0.05 m (Figure 3c).

Three-hourly MFWAM-modelled wave data were used to characterise the wave conditions during the 1 July event and compared to the wave conditions for the full year 2022 to provide context (Figure 4). The 1 July event was the most energetic wave condition that occurred in 2022 (Figure 4a) and the offshore wave conditions (at 1,000 m depth) during the peak of the event were characterised by $H_{s,o}$ = 3.30 m (Figure 4a), $T_p$ = 20.5 s (Figure 4b) and wave direction = 200° (Figure 4c).

The full MFWAM wave record (1993–2022) was also used to place the 1 July event in a longer-term perspective. The 2-, 10- and 100-year significant wave height, computed using the GVE distribution, is 2.86, 3.27 and 3.58 m, respectively, and the peak wave height during the July event of $H_{s,o}$ = 3.30 m thus has a 10-year return period. The 30-year wave record was also used to identify all major storm events that occurred over this period. Storms were identified as having a peak wave height larger than the 99% exceedance wave height, which was 2.44 m, and storms had to last at least 6 h and storm peaks had to be more than 8 h apart to count as separate events. A total of 158 storms were identified, and the date/time of each storm peak was used to extract the high tide level associated with the storm from the Gan tide record (maximum high tide (HT) level that occurred within <7 h from the time of the peak in $H_s$). Each storm was, thus, uniquely identified by an MFWAM-modelled peak storm wave height $H_{s,o}$ and peak wave period $T_p$, and measured high tide $wl$ and plotted in a $wl$-$H_{s,o}$ parameter space, with $T_p$ represented by the size and the colour of the symbols (Figure 5). This analysis indicates that the 1 July event was the third-most energetic event in terms of peak wave height and the second largest event in terms of peak wave power. The two events with the largest peak wave height over the 30-year period occurred on 5 November 2016 and 18 May 2020, with maximum $H_{s,o}$ of 3.55 and 3.52 m, but lower high tide levels of 0.24 and 0.33 m MSL, respectively, compared to 0.53 m MSL for the 1 July event. These two more energetic events in terms of peak wave height had much shorter peak wave periods: $T_p$ < 10 s, compared to $T_p$ = 20.5 s during the 1 July event.

### Impact of 1 July event

The most significant flooding occurred during the afternoon high tide on 1 July. According to the National Disaster Management Authority of Maldives, 20 inhabited islands were affected across the southern atolls of Thaa, Huvadhoo and Addu Atoll (https://time sofaddu.com/2022/07/02/sea-swells-flood-over-20-maldives-islands-causing-major-damage/).

A field team carried out visual observations and mapped the flood extent through recording the position of wrack lines and overwash deposits on the inhabited islands of Rathafandhoo, Fiyoaree and Fares-Maathoda, located along the SW rim of Huvadhoo Atoll, a few days after the event (Figure 6). Additional information on flood extent was obtained from local residents. Visual observations were also made along the western side of the uninhabited island of Dighelaabadhoo, east of Fioyaree. Evidence of island flooding and

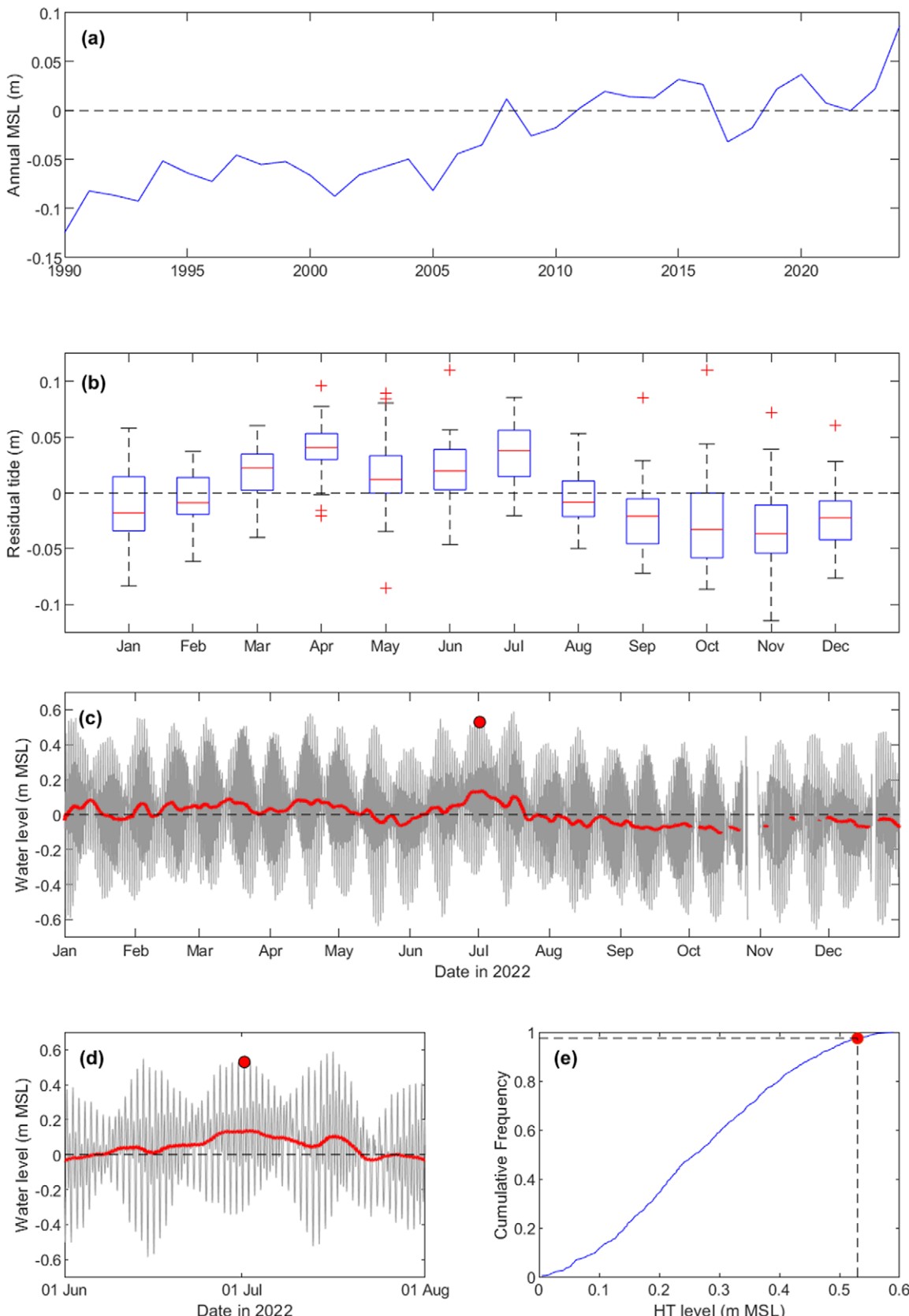

**Figure 3.** Analysis of tide gauge data from Gan: (a) Annual mean water level for period 1990–2023; (b) box plot of the monthly mean water level for the period 1990–2023; (c) water-level time series for 2022; (c) water-level time series for June and July 2022; and (d) cumulative frequency of all high tides during 2022. The 1 July event is indicated by a red circle, and the red line in panels (c) and (d) represents a 3-day moving average of the non-tidal component of the water level (i.e., residual tide). In the boxplot, the central mark indicates the median; the bottom and top edges of the box indicate the 25th and 75th percentiles, respectively; the whiskers extend to the most extreme data points not considered outliers; and the outliers are plotted individually using the '+' marker symbol.

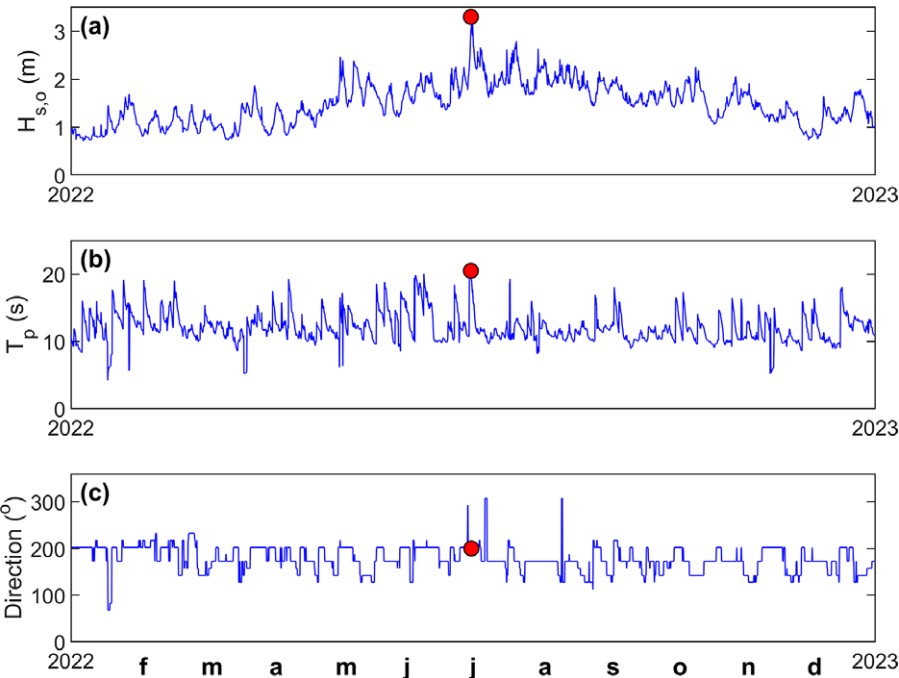

**Figure 4.** Time series of MFWAM-modelled deep-water (1,000 m depth) wave conditions for grid cell SW of Fioyaree during 2022: (a) Wave height $H_{s,o}$; (b) peak wave period $T_p$; and (c) wave direction. The 1 July event is indicated by a red circle.

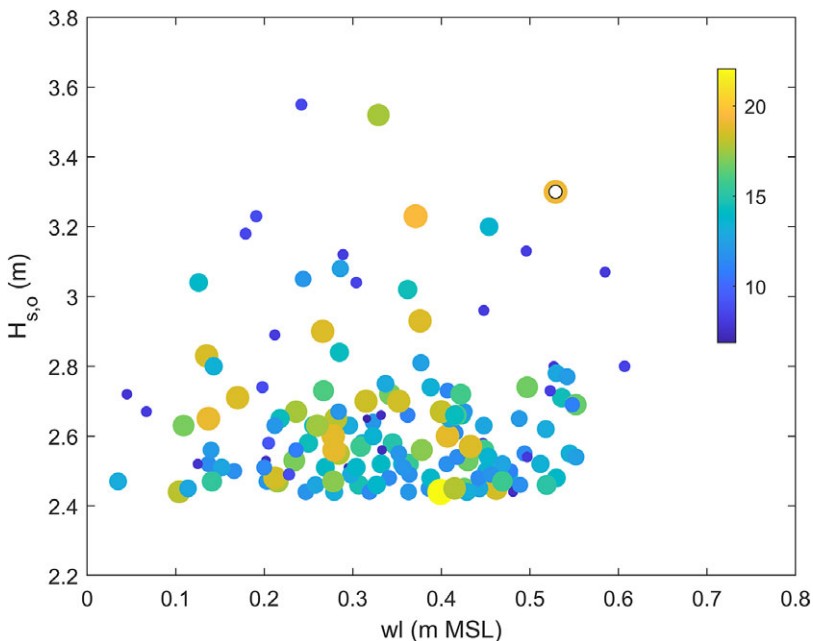

**Figure 5.** Scatter plot of offshore peak wave height $H_{s,o}$ versus associated high tide *wl* for 158 storm events identified from the offshore wave record for the period 1990–2023. The wave height was derived from the time series of MFWAM-modelled deep-water (1,000 m depth) wave conditions for grid cell SW of Fioyaree, and the water level was extracted from the Gan tide gauge record. The water level was normalised by setting the mean water level for 2022 to 0. The symbols are scaled and coloured based on the peak wave period $T_p$ at the height of the storm. The 1 July event is indicated by the yellow-orange symbol with a small white circle inside.

overwash was ubiquitous, although some of the breaches in the protective gravel ridge, typically present along most of the island margin, had already been repaired. Of the islands surveyed, Rathafandhoo was the most affected with fresh overwash deposits of variable thickness (0.05–0.2 m) covering a 10–30-m-wide strip of the low-lying SE part of the island and encroaching on the mangroves (Figure 7a,b,c). On Rathafandhoo, about 25% of the land area was flooded (Figure 6a) and 90 properties were affected with maximum flood depths of c. 0.2 m. On Fiyoaree and Fares-Maathoda, only a handful of properties were affected by flooding, and here, overwashing was most extensive along the SW part of the islands, especially on Fiyoaree (Figure 6b,c).

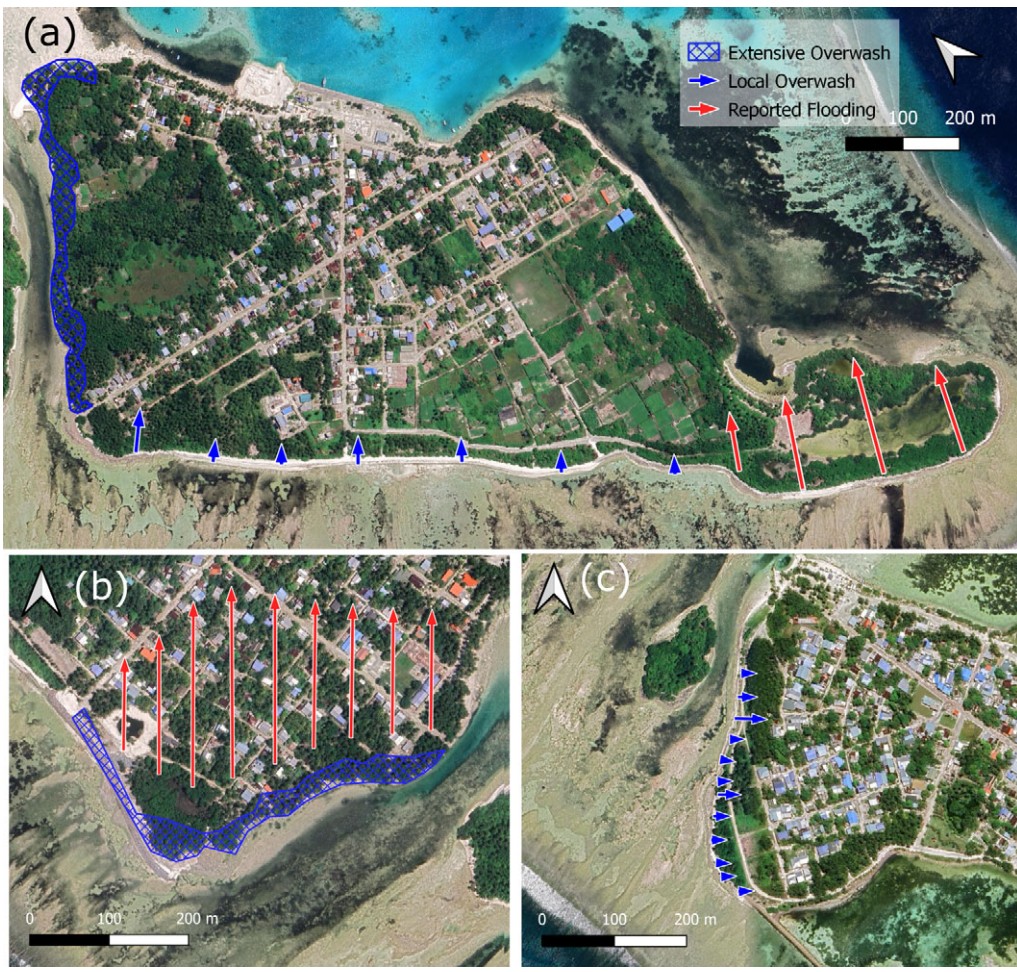

**Figure 6.** Mapped extent of island flooding and overwash deposits recorded after 1 July flooding event: (a) Fioyaree; (b) SE margin of Rathafandhoo; and (c) SW margin of Fares-Maathoda. See Figure 1c for the location of the three atoll islands. Source: GoogleMap.

Flooding pathways were generally breaches or depressions (due to public access) in the natural and anthropogenically enhanced gravel ridge (Figure 7f), and then by means of roads and tracks conveying flood water to the low-lying island interior. The extent of the washover deposits, whether sand or coral rubble (Figure 7d,e), appeared to be very much related to the density of the vegetation, with a densely vegetated island margin limiting the landward extent of the washover deposits to <5–10 m. Along the more coarsely grained, open-coast sections of the islands, some of the coral rubble ridges experienced construction (Figure 7h), as demonstrated by the presence of coral rubble that was freshly sourced from the conglomerate platform present along part of the ocean margin of the SW Huvadhoo islands (Figure 7g). Comparison of a profile measured across the gravel part of Dighelaabadhoo, between May 2022 (provided by Paul Kench, pers. comm.) and July 2022, showed that the ridge had accreted by up to 0.5 m. Video footage during the flood event on Fares-Maathoda (Figure 7i) shows that overwashing waves had maximum depths of at least 0.2 m.

## Methods

### Field campaigns

Two field campaigns were held on Fiyoaree in 2022: a 2-week campaign in January, followed by a 1-week campaign in July that took place only a few days after the 1 July island flooding event. During the first campaign, topographic, bathymetric and ecological surveys were conducted and several sediment samples were collected from the ocean and lagoon beach and from sediment cores. Island and reef platform topography was obtained using a combination of Real-Time Kinematic Global Navigation Satellite System (RTK-GNSS), automatic level, Uncrewed Aerial Vehicle and boat-based single-beam echosounder. All survey data were related to MSL using several months of sea level recorded in the Fares-Maathoda harbour in 2022 using a tide gauge (vented pressure sensor). During the second field campaign, 10 pressure sensors (PTs) were deployed (Figure 8) over 2 months during the generally more energetic July–August period, and data were collected continuously at 2 Hz. The extent of overwash deposits and island flooding resulting from the 1 July event was mapped.

Instrument positions and elevations of the PTs on the reef platform and in the harbour were recorded using RTK-GNNS (with error $dz = 0.03$ m) and related to MSL using the tide gauge data. Determining the offshore water level from a submerged PT is challenging as the elevation of the sensor cannot readily be measured. Here, the elevation of the submerged PT $z$ was obtained by computing the mean water *depth* over the field campaign $<h>$ and relating this to the mean water *level* in the lagoon $<\eta>$ (recorded by the harbour PT): $z = <\eta> − <h>$. This approach is justified because the ocean and lagoon tidal signal at the field site are very similar

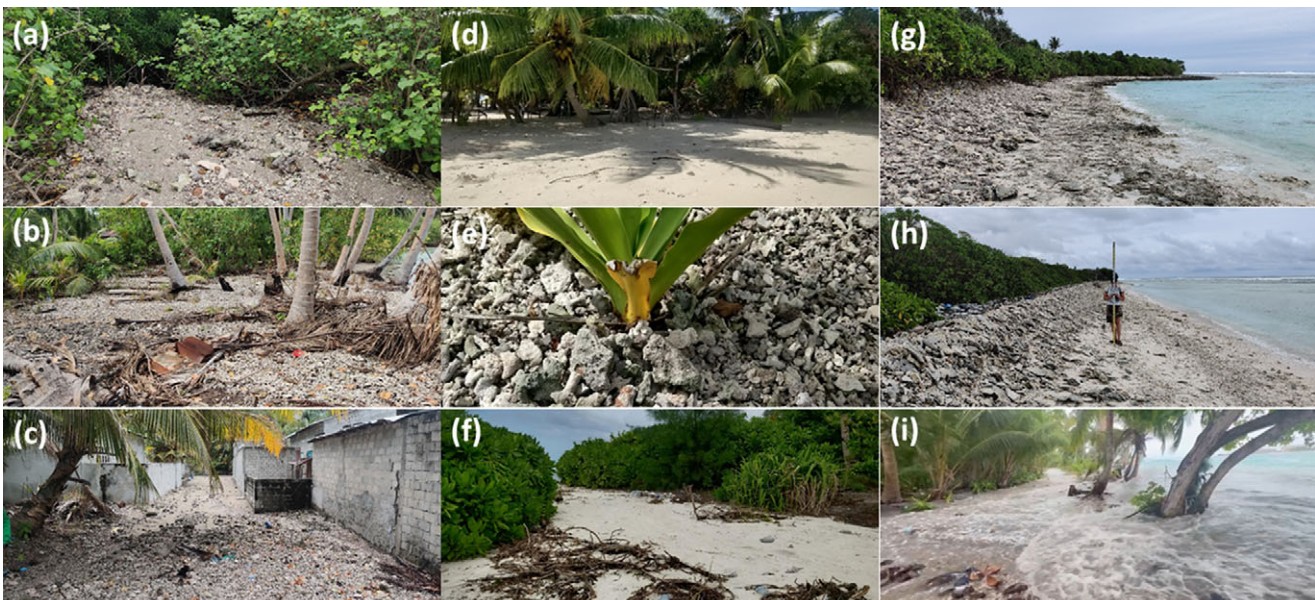

**Figure 7.** Photos taken within several days of the 1 July flooding event. Extensive, predominantly gravel, overwash deposits on Rathafandhoo into the (a) mangrove and (b) palm forest and (c) into the village via the roads. Mixture of (d) sandy and (e) gravel overwash sheets on Fiyoaree, and where a gravel ridge is present, (f) overwash tongues are found where there are gaps in the vegetation and/or the ridge. (g) Along the ocean side of most atoll islands in South Huvadhoo atoll, including on Fiyoaree, a conglomerate platform is present, and, on many locations, large pieces of coral rubble were mined from this platform and piled onto or over the gravel ridge. (h) Along the SW margin of Dighelaabadhoo, the existing gravel ridge was raised by 0.5 m and steepened to a 1:1 slope. (i) Footage taken during the flooding event on Fares-Maathoda shows overwashing waves with depths of 0.2–0.5 m, leading to flooding along the SW margin of the island. Source: Authors.

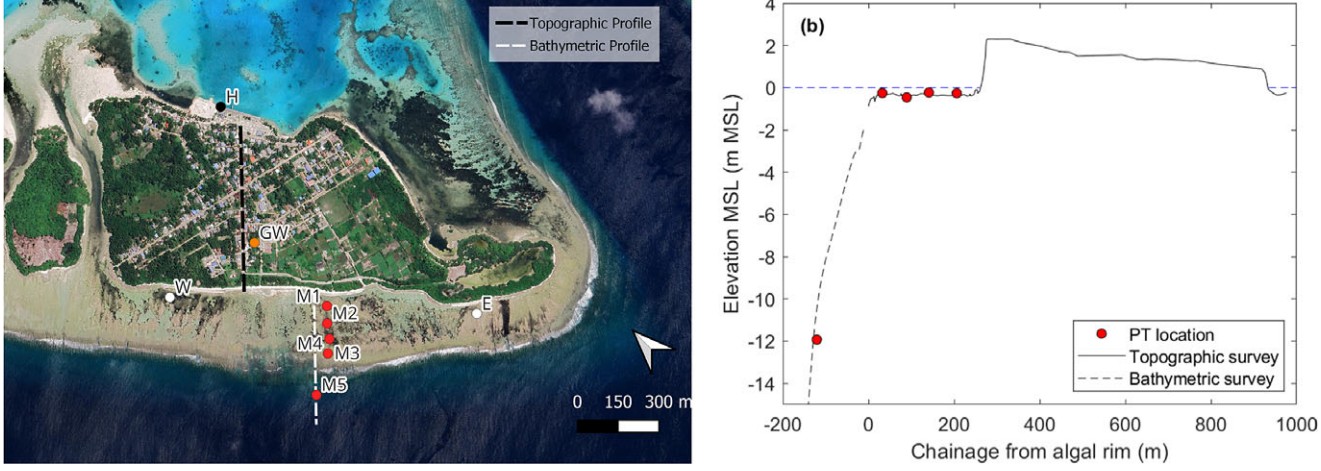

**Figure 8.** (a) Satellite image of Fioyaree with location of all instruments during the July 2022 field campaign; and (b) assembled cross-island-platform profile derived from combined topographic (RTK-GNSS and electronic level) and bathymetric survey (single-beam echosounder) with the locations of the wave sensors on the central transect. Source of (a): GoogleMap.

(Supplementary Figure S1a). High tide in the lagoon and the ocean are concurrent, and the water level difference between the lagoon and ocean never exceeds 0.1 m, with a root mean square error (RMSE) of 0.03 m (Supplementary Figure S1b). Scatter plots between the ocean and lagoon water levels further reveal that there is no systematic tidal dependency in the difference between the respective water levels (Supplementary Figure S1c,d). Any small differences between the lagoon and ocean water levels can easily be explained in terms of wind setup (especially in the lagoon).

Hydrodynamic data analysis focused on 1-h time series to represent a compromise between tidal non-stationarity and the need to have sufficiently long time series to study infragravity

(IG; $f$ = 0.005–0.05 Hz) and very low frequency (VLF; $f$ = 0.001–0.005 Hz) wave energy. Sea-swell wave energy (SS) is defined by wave motion at frequencies <0.05 Hz. All wave heights are significant wave heights and were computed as four times the square root of the spectral variance summed across the relevant frequency bands. During low tide conditions, water depths across the reef platform were generally <0.25 m, and ponding occurred due to the raised algal rim present along most of the seaward margin of the reef platform. Therefore, the hydrodynamic analysis presented here focused on mid- to high tide conditions, referred to as 'HT half cycles', for which the offshore water level is >0 m MSL. The offshore wave data, collected in 12 m water depth, was corrected for depth

attenuation and deshoaled to 25 m using linear theory, and is denoted by $H_{s,25m}$.

## Numerical modelling

The phase-resolving version of XBeach (XB-NH) was used for the numerical modelling because, compared to the phase-averaged (XB-SB) model, this version performs better for run-up on steep beaches (McCall et al., 2014; De Beer et al., 2021) and in coral reef environments (Quataert et al., 2020), and also has fewer tuning parameters because the physics are better represented. The model was run in one-dimension (1D) because of computational efficiency and the lack of a complete digital elevation model of the island. A central profile line was extracted from the survey data and included the transect with five PTs (Figure 8). Numerical modelling was only conducted using this central transect.

A large number of model calibration runs were conducted using a subset of data comprising all 1-h time series for the period 10–16 July 2022 ('calibration data set') for which the ocean water level was >0 m MSL ($N = 86$). All model runs were forced with the measured water level and a JONSWAP wave spectrum characterised by $H_{s,25m}$ and $T_p$ from the offshore PT and the offshore model boundary was at 25 m depth. The sole calibration parameter was the bed roughness, quantified by the Colebrook-White roughness $cf$ in XB-NH, and this parameter was considered spatially variable, with the largest roughness attributed to the forereef with spur-and-groove topography. In the model, the subtidal region and the reef platform were considered impermeable, whereas the island was considered permeable with a hydraulic conductivity of $K = 0.005$ ms$^{-1}$. Model performance was evaluated using the RMSE, bias and correlation coefficient ($r$). The variables for which the model was calibrated were extracted from the PT closest to the island (M1) and included wave setup $\eta$ and the total, SS, IG and VLF wave height ($H_s$, $H_{s,SS}$, $H_{s,IG}$ and $H_{s,VLF}$, respectively).

The objective of this research is to investigate the 1 July 2022 flooding event, and, ideally, wave run-up should be used for model calibration as wave run-up elevation in relation to island elevation controls overwash and flooding. It is important on the outside to distinguish between the run-up *height* (vertical distance of run-up above still water level [SWL]) and the run-up *elevation* (vertical distance of run-up above the MSL datum). Run-up observations were, however, not available, and the total water height (TWH), defined here as ($\eta + H_{s,SS} + H_{s,IG} + H_{s,VLF}$), was used for calibration as a surrogate for run-up height. It is noted that this simple linear addition of wave terms is not physically correct, but it was found that the maximum run-up height was very well predicted by the modelled TWH at the toe of the island beach. After model calibration, the model was validated using a different subset of the hydrodynamic data for the period 18–31 July 2022 ('validation data set'), also for which the ocean water level was >0 m MSL ($N = 164$).

The validated/calibrated XBeach model was used to simulate the 1 July flooding event, forced by the Gan water level data and the MFWAM wave data (with a wave height correction to account for nearshore wave transformation). The modelled run-up elevation during the event was compared to the elevation of the island and to visual observations of island flooding.

The calibrated and validated XBeach-NH model was also used to develop a run-up predictor for Fioyaree based on offshore wave conditions and water level. A large number ($N = 231$) of simulations were conducted using a combination of offshore wave height ($H_{s,o} = 2$–4 m) and water level ($wl = 0$–1 m MSL). Initially, a peak wave period of $T_p = 20$ s was assumed in these simulations, as such a period is commonly associated with island flooding in the Maldives

(Wadey et al., 2017). For model forcing, the offshore wave height was corrected to account for nearshore wave transformation, and the model was forced with $H_{s,25m}$. The measured profile used in the previous simulations was modified to create a beach face that extends to 3 m MSL to focus only on run-up and not overwash. A hydraulic conductivity value of $K = 0.005$ ms$^{-1}$ and the calibrated spatial variability in the bed roughness ($cf = 0.15$ in the spur-and-groove region and $cf = 0.01$ elsewhere) were used. The 2% exceedance run-up $R_{2\%}$ elevation, rather than the maximum run-up $R_{max}$ elevation, was extracted from the modelled data set and visualised within the $wl$-$H_{s,o}$ parameter space as a nomogram. The effect of changing sea level on these results was then evaluated.

The role of $T_p$ on run-up, and thus potential for overwash and island flooding, was also investigated. Using a constant water level of 0.53 m MSL and a constant offshore wave height of 3 m (as during the high tide peak of the 1 July event), XBeach was forced with $T_p$ ranging from 5 to 25 s and $R_{2\%}$ was investigated. Model parameters were the same as for the runs used for constructing the nomogram. Finally, the importance of $T_p$ was also addressed by computing $R_{2\%}$ for all the storm events that occurred over the 1990–2023 period. XBeach was forced by the maximum $wl$, $H_{s,o}$ and $T_p$ that occurred during each event. Because the $wl$ value for the identified storms reflects the long-term change in sea level at Gan (*cf.*, Figure 3), $wl$ associated with each storm was recalculated from the detrended Gan sea-level record and was normalised by setting the mean sea level for 2022 to 0.

## Results field campaign

### Morphology and sedimentology

A single representative cross-island and cross-reef profile was extracted from the survey data (Figure 8b). This profile represents a blending of the profile across the island via the main road (using automatic level), the transect across the reef platform with the main instrument array (using GNSS) and a subtidal profile (using echosounder). The transect was also used in the XBeach modelling. At the location of the transect, the island is 650 m wide, and the crest elevation at the lagoon and ocean side is 0.9 m MSL and 2.3 m MSL, respectively. The reef platform is 220-m wide and has an average elevation of $-0.4$ m MSL. At its ocean side, there is an algal rim with an elevation of $-0.15$ m MSL. The algal rim slopes gently down from its highest elevation over a distance of c. 30 m, and then the forereef steeply slopes down with a gradient of 0.08.

The beach at the ocean-side of the island is composed of predominantly medium-to-coarse sand with a $D_{50}$ at the West, Main and East transect of 0.56, 0.33 and 0.67 mm, respectively, and has a steep gradient of 0.1–0.15. The back of the beach is generally characterised by a coral rubble ridge with varying clast sizes (1–10 cm). The surface sediments of the island are highly variable, and size analysis of shallow (0.3 m) sediment cores taken from across the island shows a $D_{50}$ value ranging from 0.15 to 13.1 mm and half of the core samples are characterised by a bimodal sediment distribution containing sand and gravel. The beach at the lagoon-side of the island tends to consist of finer sediments than the ocean-side beach, and analysis of a single sample yielded a $D_{50}$ of 0.31 mm.

### Hydrodynamics

The July–August 2022 field deployment covered almost three complete neap-to-spring tidal cycles (Figure 9a) and wave heights $H_{s,25m}$ in excess of 1.5 m with peak wave periods $T_p$ of 15–18 s were

experienced (Figure 9b,c). Normalised wave spectra suggest that these energetic episodes were remote swell events (as opposed to wind-wave events), as the peak wave period decreased (as opposed to increased) during the events (Figure 9d), reflecting wave dispersion (as opposed to wave generation). Comparison between the measured wave data (deshoaled to 25 m depth) collected over a 6-week period (9 July–17 August 2022) and the overlapping MFWAM-modelled wave data (shoaled to 25 m depth) indicates that the measured $H_{s,25m}$ is 63% of the modelled $H_{s,25m}$. The peak wave period is poorly predicted by MFWAM ($r = 0.31$, but no obvious bias; not shown); therefore, the modelled $T_p$ without any correction is used for the 1 July flood event simulation.

The energetic period 18–31 July 2022 was selected for investigating the cross-reef variation in hydrodynamics, specifically the mean water level (wave setup) and wave conditions. Only HT half cycles were considered. The across-reef variation in wave set-up $\eta$, total wave height $H_s$ and combined IG and VLF wave height $H_{s,IGVLF}$ is illustrated in Figure 10a–c for three examples 1-h data segments. None of the time-averaged hydrodynamic parameters show much variability across the reef platform (*cf.* Supplementary Table S1), but $\eta$ and $H_{s,IGVLF}$ do show a strong dependence on the offshore wave conditions, with the largest and smallest values associated with the largest and smallest $H_{s,25m}$, respectively. This is further explored in Figure 10d–f, which represents scatter plots of $H_{s,25m}$ versus the hydrodynamic parameters averaged across the reef platform with their best-fit lines. A strong linear dependence of $\eta$ and $H_{s,IGVLF}$ on $H_{s,25m}$ is apparent with *r*-values of 0.86 and 0.62, respectively. As $H_{s,SS}$ across the reef is not significantly related to $H_{s,25m}$, and is mainly related to the water depth across the reef platform (not shown), the significant correlation between $H_{s,25m}$

and $H_s$ across the reef platform ($r = 0.50$) is mainly attributed to the relation between $H_{s,25m}$ and $H_{s,IGVLF}$. Figure 10d,f suggest that $\eta$ and $H_{s,IGVLF}$ across the reef platform are roughly 10–20% of $H_{s,25m}$, which is in line with observations on other coral reefs (e.g., Vetter et al., 2010; Pomeroy et al., 2012).

The same subset of data collected over the energetic period 18–31 July 2022 was subjected to further time- and frequency-domain analysis to investigate the mechanism for generating the combined IG and VLF motion across the reef platform. Wave spectra computed for data collected at the base of the beach show a prominent SS and IG wave peak at 13–14 and 25–30 s, respectively, with a clear spectral value at a frequency of 0.05 Hz (Supplementary Figure S2). Most wave energy is contained within the IG domain (20–200 s), but significant amounts of wave energy are also present within the VLF domain (>200 s). The spectrum associated with HT half cycles also displays a broad peak at 4–5 s, associated with secondary waves that typically develop over horizontal surfaces (Masselink, 1998).

Following List (1992), further analysis was conducted between the offshore wave groupiness and the low-frequency water motion across the reef platform. The raw offshore wave signal (uncorrected for depth attenuation) was used to obtain the wave groupiness signal $GF$ by taking the modulus of the detrended time series and low-pass filtering using a simple Fourier filter with a cut-off frequency of 0.02 Hz. The wave signal across the reef platform was simply low-pass-filtered using the same 0.02 Hz cut-off. The different low-pass-filtered signals can be compared visually (Supplementary Figure S3a), but cross-correlation analysis was conducted to quantify maximum correlation coefficients and associated time lags (Supplementary Figure S3b,c).

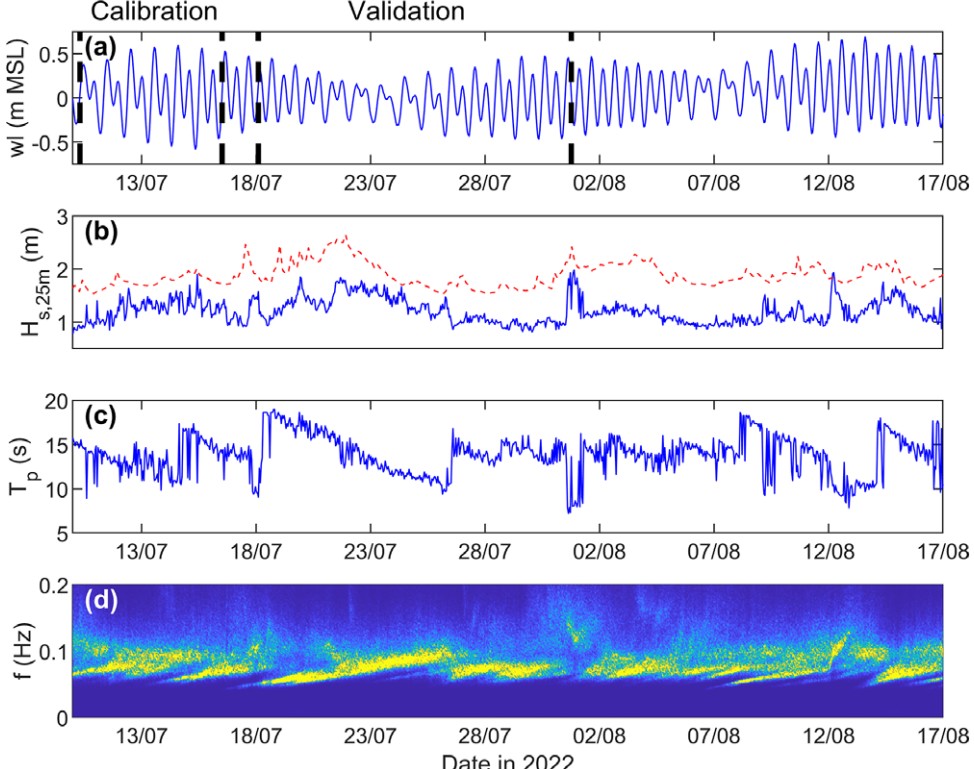

**Figure 9.** Summary of the hydrodynamic conditions during the July 2022 field campaign. Time series of: (a) detrended water level *wl*, (b) measured (blue line) and MFWAM-modelled (red dashed line) wave height in 25 m water depth $H_{s,25m}$ (c) measured peak wave period $T_p$, and (d) measured normalised wave spectra (spectral energy increases from dark blue to yellow). Measured wave conditions are based on data collected by a pressure sensor deployed in c. 12 m water depth, with the wave height deshoaled to 25 m using linear theory. The modelled wave height represents the MFWAM-modelled offshore wave height shoaled to 25 m using linear theory.

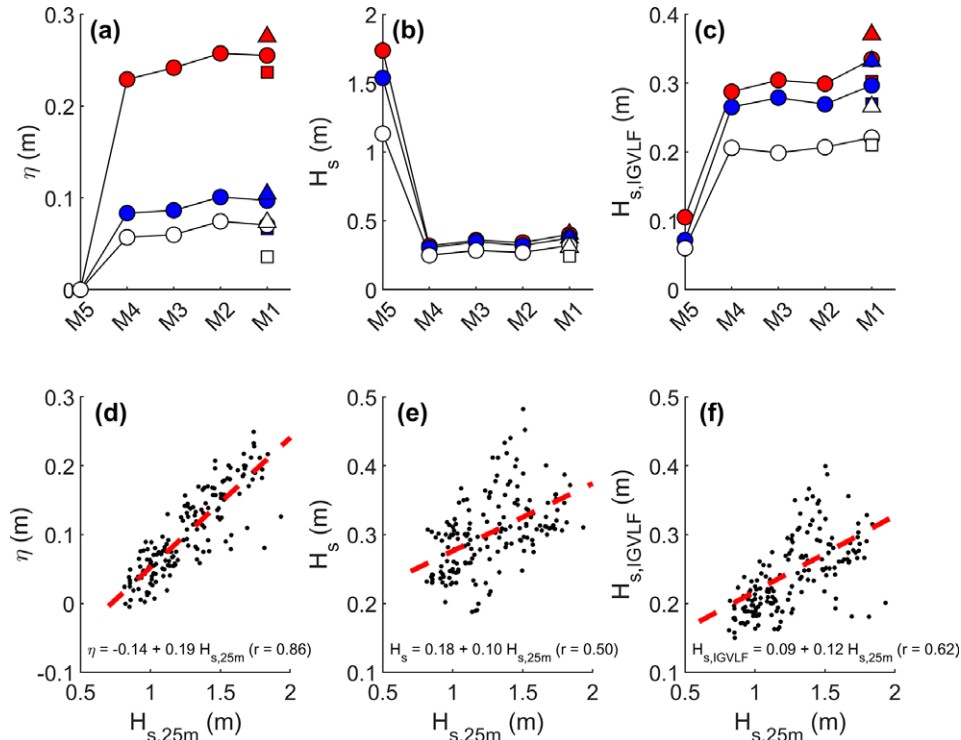

**Figure 10.** Example of across-reef variation, from M5 (offshore) to M1 (toe of island beach), in (a) wave set-up $\eta$, (b) wave height $H_s$ and (c) IG and VLF wave height $H_{s,IGVLF}$ for a 1-h data segment with wave height $H_{s,25m}$ of 1.74, 1.54 and 1.13 m (red, blue and white symbols, respectively). The wave statistics for E (triangles) and W (squares) are also plotted. Scatter plot and best-fit lines of across-reef averaged (d) $\eta$, (e) $H_s$ and (f) $H_{s,IGVLF}$ taking into account all 1-h pressure sensor data segments collected on the reef (M1–M4, E and W) for the period 18–31 July 2022, and only considering the top part of the tidal cycles (offshore water level $wl > 0$ m).

The overall picture that emerges from this analysis is that of a breakpoint-forced wave that is generated across the forereef in the breaking zone and then propagates onshore across the reef platform as a free wave at the shallow water wave speed (Pomeroy et al., 2012; Masselink et al., 2019).

## Results of numerical modelling

### Modelling calibration

In the phase-resolving XBeach model, the bed roughness *cf* is the main calibration parameter, and calibration involved varying *cf* and investigating the mean RMSE, bias and *r* computed considering all 1-h time series collected over the period 10–16 July 2020 for which the ocean water level was >0 m MSL ($N = 86$). Of the three model performance indicators, the bias is considered the most relevant indicator because, due to the relatively large correlation values ($r > 0.8$), the RMSE is mainly determined by the bias. The model was relatively insensitive to the bed roughness in the subtidal region beyond 10 m water depth and across the platform and beach, but was extremely sensitive to the bed roughness across the spur-and-groove section (from $-10$ m MSL to the algal rim). The calibration results with enhanced roughness over the spur-and-groove section are illustrated in Figure 11a and tabulated in Supplementary Table S2. The bed roughness for the island, the reef platform and the deeper subtidal region was set to $cf = 0.01$ in these calibration runs.

A low bed roughness ($cf = 0.01$; relatively smooth surface) across the entire model domain results in a large over-prediction of wave setup and wave height across the reef platform (bias >0.15 m). A large roughness ($cf = 0.3$; highly rugose surface) across the spur-

and-groove section only modestly reduces the over-prediction of setup (bias <0.1 m), but results in a large under-prediction in the IG and VLF wave height (bias < $-0.15$ m) (Figure 11a and Supplementary Table S2). It is impossible to accurately model both wave setup $\eta$ and IG and VLF wave height simultaneously: if good agreement in $\eta$ is reached, the wave heights are under-predicted, whereas if good agreement in wave height is reached, $\eta$ is over-predicted. This overprediction of setup (or underestimation of IG and VLF waves at the expense of correct setup predictions) has previously been shown by Van Dongeren et al. (2013) to be a limitation of 1D modelling in cases where shoreward mass flux at the reef crest is transported alongshore on the reef platform or lagoon, which is not accounted for in a 1D model. The only way in the 1D model to bring the modelled wave setup in line with the observations, whilst not overly reducing the $H_{s,IGVLF}$, is by reducing the gradient of the spur-and-groove section (not shown); however, this is clearly inappropriate as the subtidal gradient is a measured boundary condition and not a calibration parameter.

Our main interest is in correctly modelling wave run-up, because the elevation of the run-up compared to island elevation determines island freeboard (Matias et al., 2012), which in turn controls occurrence and magnitude of overwash. However, there are no field observations of run-up to compare the model results with. The run-up height in the model can be parameterised by the TWH at the toe of the island beach, defined as ($\eta + H_{s,SS} + H_{s,IG} + H_{s,VLF}$) (Figure 11b). Therefore, the XBeach model was optimised by aiming for good agreement with the observed TWH, while accepting that this will overestimate $\eta$ and underestimate $H_{s,IG}$ and $H_{s,VLF}$. The closest agreement between modelled and measured TWH is achieved with a bed roughness of $cf = 0.15$ across the spur-and-groove region (Figure 11a and Supplementary Table S2), which is in

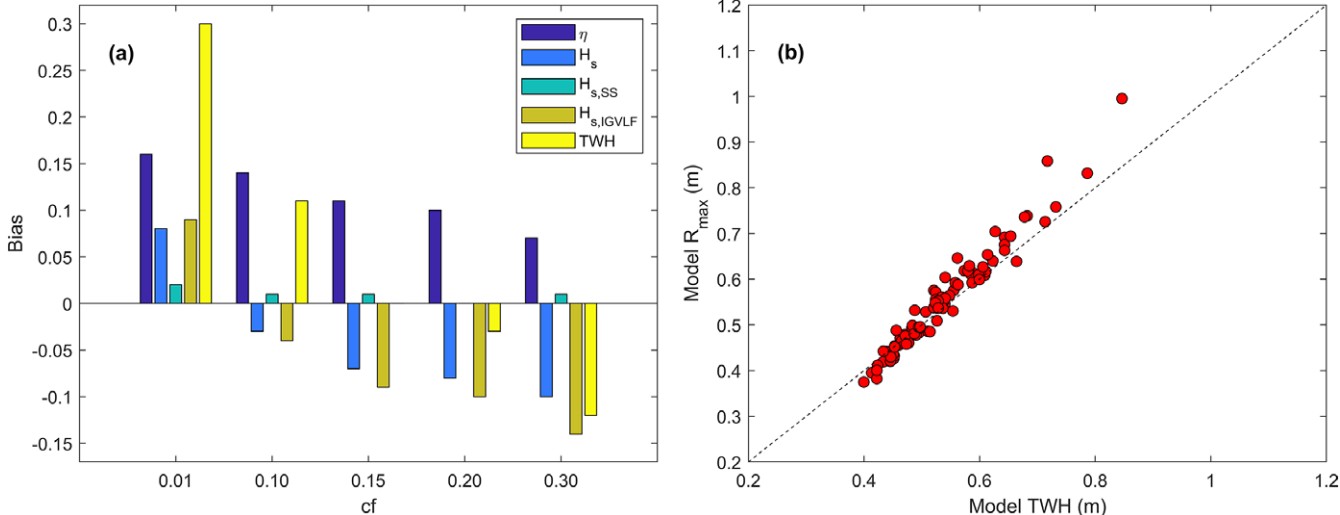

**Figure 11.** (a) Model bias averaged over the calibration data set for the different hydrodynamic components as a function of bd roughness *cf*. The hydrodynamic components are for the innermost PT (M1) and include: wave setup $\eta$, total wave height $H_s$, SS wave height $H_{s,SS}$, IG wave height $H_{s,IG}$, VLF wave height $H_{s,VLF}$ and the total water height TWH (= $\eta$ + $H_{s,SS}$ + $H_{s,IG}$ + $H_{s,VLF}$). (b) Scatter plot of model TWH versus modelled maximum wave run-up height $R_{max}$. The 1:1 dashed line (b) is not the best line of fit, but used to indicate similarity between $R_{max}$ and TWH.

line with earlier studies representing forereefs with high coral coverage (e.g., Storlazzi et al., 2019; Roelvink et al., 2021), and this setting was used in all subsequent simulations. The full calibration results for the calibrated model, with *cf* = 0.15 across the spur-and-groove region and *cf* = 0.01 elsewhere, is illustrated in Supplementary Figure S4. The data are presented in conventional scatter plots of measured versus predicted parameter values and indicates overestimation of $\eta$, underestimation of $H_{s,IG}$ and $H_{s,VLF}$ and good model performance for $H_{s,SS}$ and TWH.

### Model validation

Model validation was conducted using 1-h time series for the period 18–31 July 2020, again for ocean water level *wl* > 0 m MSL ($N$ = 164). The correlation between $H_{s,25m}$ and TWH for the model results is almost identical to that for field measurements (Figure 12a,b), and the RMSE, bias and *r* associated with the comparison between measured and modelled TWH are 0.07 m, 0.06 m and 0.93, respectively (Supplementary Table S3). As TWH is considered the most relevant parameter for run-up and overwash, this is encouraging. However, model performance for the individual hydrodynamic parameters is less favourable, and $\eta$ is over-predicted, while $H_s$ and $H_{s,IGVLF}$ are under-predicted (Supplementary Table S3). The measured and modelled data were partitioned into the different hydrodynamic components, and their statistical distributions were compared (Figure 12c,d). This comparison further highlights the shortcomings in the numerical model: $\eta$ is over-predicted, while $H_{s,IG}$ and $H_{s,VLF}$ are under-predicted (Figure 12c,d). In general, the performance statistics for the validation are not as good as for the calibration. This is because the validation data set includes more energetic wave conditions (Supplementary Figure S9), and $H_{s,IG}$ and $H_{s,VLF}$ are especially under-predicted for more energetic wave conditions (Supplementary Figure S4f,g).

The model validation data were also used to investigate the mechanism of IG-VLF generation and was subjected to an identical cross-correlation analysis as the field data. The analysis confirms the presence of a group-bound long wave beyond the wave breakpoint, and that the combined IG-VLF motion across

the reef platform originates as a breakpoint-forced wave that propagates onshore across the reef platform as a free wave (-Supplementary Figure S4). The cross-correlation analysis shows very similar results for the measured and modelled data, although the amount of IG and VLF energy in the model data is significantly less than in the measured data and the cross-correlation coefficients are higher.

### Modelling the 1 July event

The calibrated and validated XBeach model was forced with the observed water-level time series from Gan and the MFWAM-modelled waves with the local correction based on measured wave data. Only the 25-h period that included two full tidal cycles with the peak of the wave event was modelled (Figure 13). Two maximum water levels over the modelled time period were extracted from the model data: one based on the maximum run-up elevation ($R_{max}$ = 1.87 m MSL) and one based on the 2% exceedance run-up elevation ($R_{2\%}$ = 1.60 m MSL). The highest water level reached during the 1 July event is made up of a 0.53 m MSL still water level, and a maximum and 2% exceedance run-up height of 1.34 and 1.07 m, respectively. The component of the run-up that is due to wave setup $\eta$ is 0.39 m.

The modelled run-up for the 1 July event cannot readily be compared to the field data due to a lack of robust run-up data. Most of the flooding during the 1 July event occurred on the SE side of Rathafandhoo (for which there are no topographic data, nor wave data) and the SW side of Fioyaree (for which there are no wave data either). Topographic data indicate that the elevation of the island ridge along the SW side of Fiyoaree is typically 1–1.5 m MSL (MoEnv, 2020), whereas the elevation of the ocean ridge is generally 2–2.3 m. However, the ocean ridge has lower sections (e.g., beach access from the island), and there were also numerous breaches in the ridge after the 1 July flood event. Island flooding is likely to have occurred via these pathways, and this notion is supported by the numerous 'tongue-shaped' overwash deposits at the back of the gravel ridge. For values for $R_{max}$ and $R_{2\%}$ of 1.87 and 1.60 m MSL, respectively, at the peak of the flooding event, widespread flooding

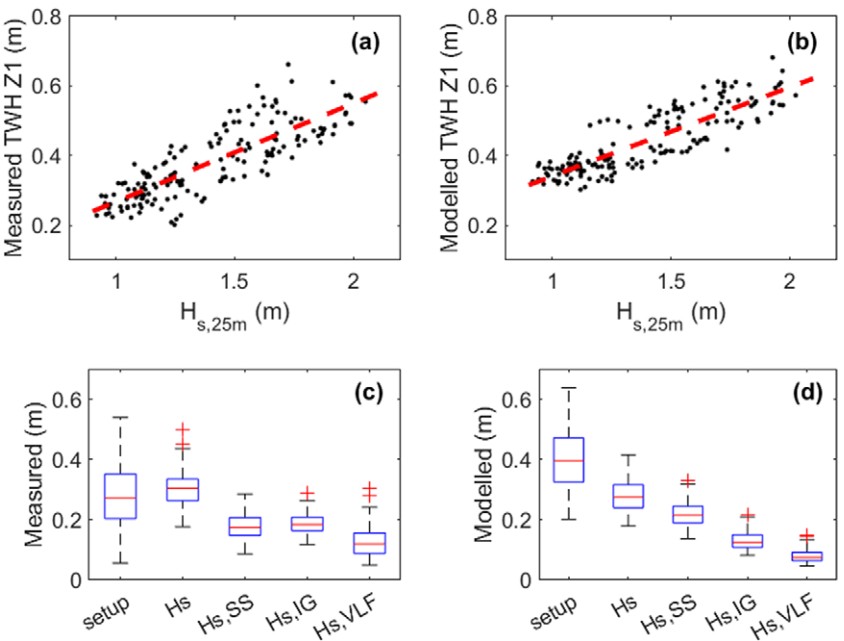

**Figure 12.** Results of XBeach model validation using data from the period 18/07 to 01/08. Scatter plot of (a) measured and (b) modelled total water height TWH (= $\eta + H_{s,SS} + H_{s,IG} + H_{s,VLF}$) at M1 versus the wave height in 25 m water depth $H_{s,25m}$. Boxplots of the different contributors to the TWH for the (c) measured and (d) modelled data. The different contributors have been averaged across the reef platform (i.e., considering M1, M2, M3 and M4): wave setup, total wave height $H_s$, SS wave height $H_{s,SS}$, IG wave height $H_{s,IG}$ and VLF wave height $H_{s,VLF}$. In the boxplots, the central mark indicates the median; the bottom and top edges of the box indicate the 25th and 75th percentiles, respectively; the whiskers extend to the most extreme data points not considered outliers; and the outliers are plotted individually using the '+' marker symbol.

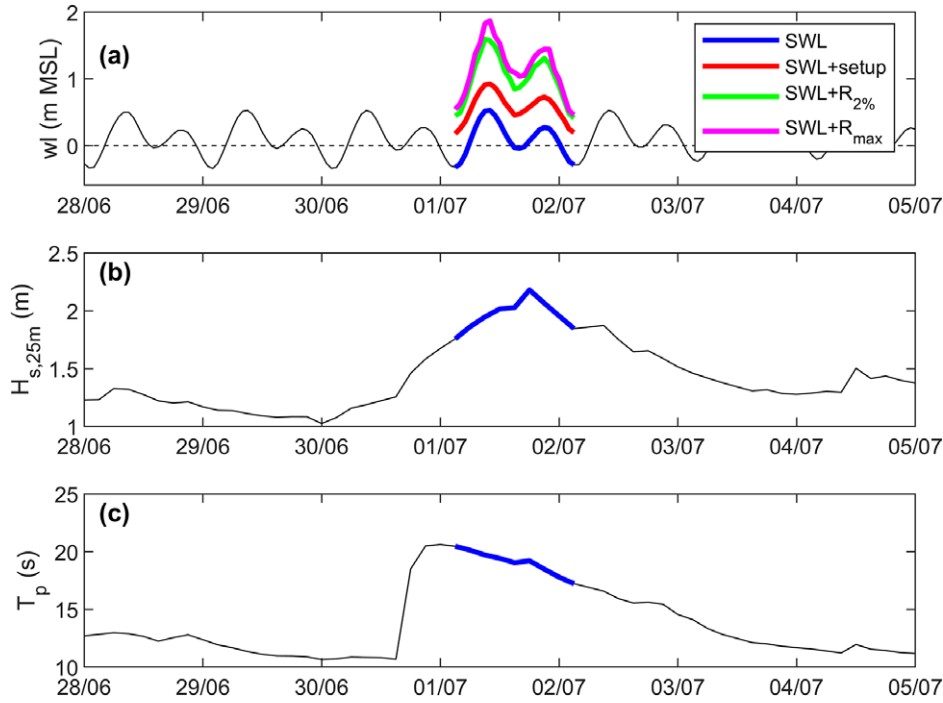

**Figure 13.** Time series of: (a) water level *wl*; (b) wave height $H_{s,25m}$; and (c) peak wave period $T_p$ for the week, including the 1 July 2022 flooding event. The blue lines represent the model boundary conditions, and the red, green and magenta lines in panel (a) are the modelled water levels. Only the 25 hours indicated by the bold lines have been modelled. The water level data are the measured tide level from Gan and was normalised by setting the mean water level for 2022 to 0.

would be expected along the SW side of Fioyaree (Figure 7d) and localised flooding associated with topographic lows in the ridge on the ocean side (Figure 7f). However, during the post-event flood mapping, there was also plenty of evidence of minor overwash where the gravel ridge seemed relatively intact. In addition, observations on Dighelaabadhoo, located just east of Fioyaree, of washover deposits at the back of the 2 m high gravel ridge suggest that run-up did significantly overwash the ridge.

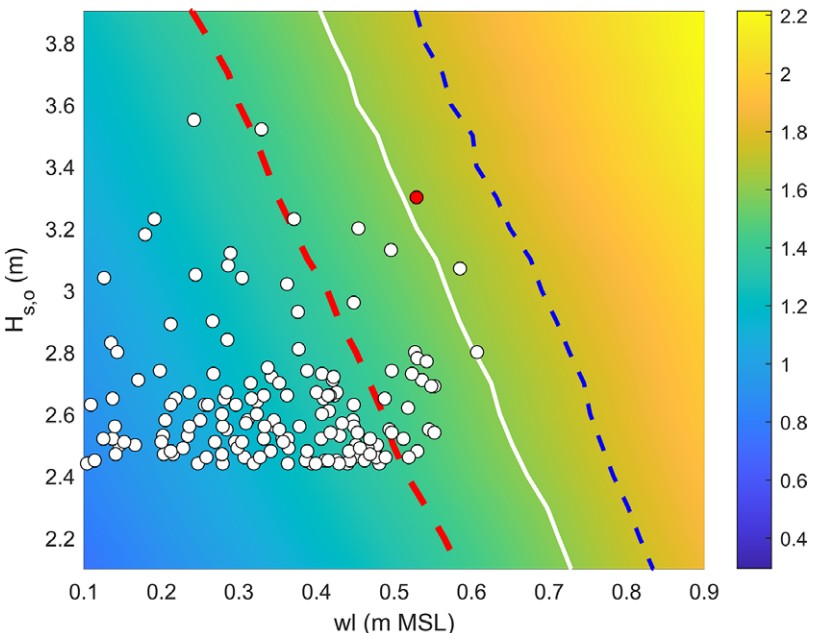

**Figure 14.** Nomogram showing the modelled 2% exceedance run-up elevation $R_{2\%}$ (relative to MSL) as a function of water level *wl* and deep water wave height $H_{s,o}$, and a constant peak wave period of $T_p = 20$ s. The circles show the 158 extreme events that occurred over the period 1990–2023, with the red circle representing the 1 July event, positioned in the nomogram based on their peak $H_{s,o}$ (MFWAM-modelled) and peak (measured from Gan tide record) *wl*. The Gan water-level record was normalised by setting the mean water level for 2022 to 0. The red dashed, white solid and blue dashed lines represent the $R_{2\%} = 1.4$, 1.6 and 1.75 m MSL contour lines, respectively.

## Application of the model to past, present and future island flooding

The calibrated and validated XBeach-NH model for Fioyaree was used to plot the 2% exceedance run-up elevation $R_{2\%}$ in a nomogram as a function of offshore water level *wl* and offshore wave height $H_{s,o}$, for a constant peak wave period of $T_p = 20$ s (Figure 14). All major storm events that occurred over the period 1993–2022, including the 1 July event, are plotted within this parameter space based on their MFWAM-modelled offshore peak wave height $H_{s,o}$ and the associated measured high tide *wl* (Figure 14). $R_{2\%}$ increases with both *wl* and $H_{s,o}$, and only the upper-right part of the nomogram is associated with overwash and island flooding ($R_{2\%} > 1.5$ m MSL). Three contour lines are visualised in the nomogram: (1) 1.6 m MSL, which represents the $R_{2\%}$ elevation associated with the 1 July event and which resulted in extensive island overwash (representing a $R_{2\%}$ *height* of 1.1 m); (2) 1.75 m MSL, which would represent the $R_{2\%}$ elevation required to cause extensive island flooding similar to the 1 July event (representing a $R_{2\%}$ *height* of 1.25 m), in 1990 when MSL was 0.15 m lower than present; and (3) 1.4 m MSL, which would represent the $R_{2\%}$ elevation required to cause extensive island flooding similar to the 1 July event (representing a $R_{2\%}$ *height* of 0.9 m), in 2050 when MSL is expected to be 0.2 m higher than present according to SSP-4.5 (IPCC, 2021).

There are three major implications of Figure 14 and these have application beyond the island of Fioyaree, as the 1 July flooding event affected 20 islands along the SW rim of Huvadhoo Atoll. First, only three events occurred over the period 1990–2023 that passed the overwash limit: 6 May 2004 pm, 15 October 2008 pm and 1 July 2022 pm. There was reported flooding in May 2004 only at one island in Huvadhoo Atoll (Hoandedhoo), and no flooding reports for the 15 October 2008 event. However, neither were swell-wave events, as the peak wave period during both storms was <15 s (Figure 5). It is, thus, suggested that significant flooding of islands

along the SW rim of Huvadhoo is currently, roughly, a 1:25-year event. Second, it is unlikely that the 1 July event would have resulted in overwash if it had occurred in 1990. Third, 25 storm events that occurred over the period 1990–2023 are plotted above the $R_{2\%} = 1.4$ m MSL contour in Figure 14 (these events would have a run-up *height* of >0.9 m) and can thus be considered capable of inducing overwash and island flooding by 2050.

However, the role of the peak wave period $T_p$ during the storm events should also be considered. It is widely accepted that island flooding in the Maldives is associated with long-period ($T_p > 15$ s) swell (Wadey et al., 2017) and the nomogram in Figure 14 was derived using a $T_p$ of 20 s, as this was the $T_p$ during the 1 July event. However, not all storm events identified from the MFWAM-modelled offshore wave record are characterised by a large $T_p$ (Figure 5). The sensitivity of wave run-up to $T_p$ was investigated by comparing the 2% exceedance run-up elevation $R_{2\%}$ predicted by the XBeach model for a constant water level *wl* = 0.53 m MSL and deep water wave height $H_{s,o} = 3.0$ m, but for varying $T_p$. The modelled runup increases with wave period (Supplementary Figure S6). The dependency of $R_{2\%}$ on $T_p$ is particularly significant for $T_p < 15$ s, with $R_{2\%}$ rapidly reducing with decreasing $T_p$.

Acknowledging the importance of wave period, therefore, XBeach was also used to compute the $R_{2\%}$ elevation for all storm events that occurred over the 1990–2023 period, using not only their individual maximum *wl* and $H_{s,o}$ but also the $T_p$ that occurred during each of the storms. Assuming an $R_{2\%}$ elevation of ≥1.4 m MSL is required to initiate extensive overwash in 2050 (when sea level is 0.2 m higher than at present), it can be concluded that 10 of the storm events that occurred over the 1990–2023 period, all with $T_p > 15$ s, would likely result in island overwash on the scale of the 1 July event (Figure 15). It is, therefore, concluded that by 2050, island flooding is likely to occur on average every few years.

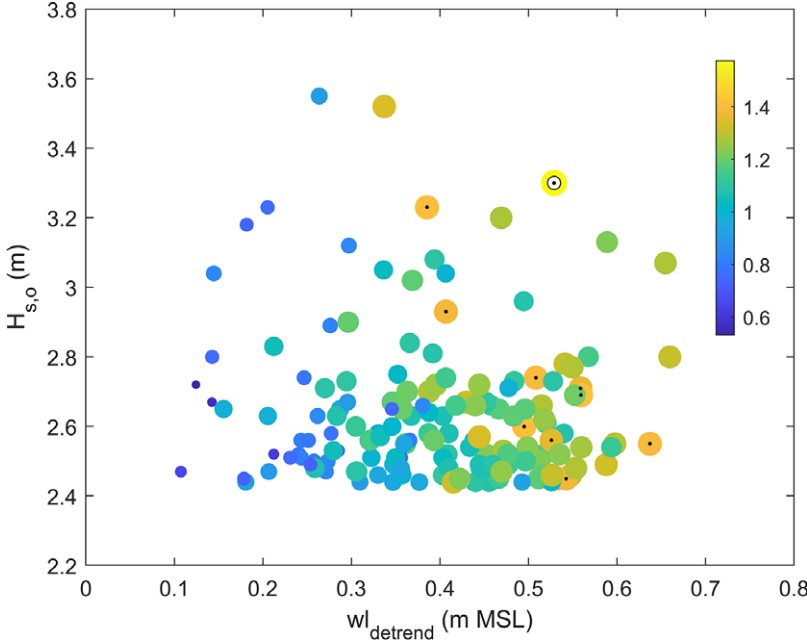

**Figure 15.** Scatter plot of offshore peak wave height $H_{s,o}$ versus associated high tide $wl$ for 158 storm events identified from the offshore wave record for the period 1990–2023. The wave height was derived from the time series of MFWAM-modelled deep-water (1,000 m depth) wave conditions for grid cell SW of Fioyaree, and the water level was extracted from the Gan tide gauge record. The water-level record was detrended, and the water level was normalised by setting the mean water level for 2022 to 0. The symbols are scaled and coloured based on the 2% exceedance run-up elevation $R_{2\%}$ (relative to MSL). The 1 July event is indicated by the yellow symbol with a small white circle inside; all symbols with a black dot represent $R_{2\%} >= 1.4$ m MSL.

## Discussion and conclusions

A large number (20) of southern Maldivian atoll islands were significantly affected by a flooding event that took place on the afternoon of 1 July 2022. Flooding was especially extensive along the SW margin of Huvadhoo Atoll, where flooding was reported to be the worst since at least the 2004 Boxing Day tsunami, making it a rare event. Significant coastal flooding of islands is actually surprisingly uncommon in the Maldives (inland flooding due to excessive rainfall is a much greater hazard) and probably stems from the fact that many Maldivian atolls were established during the mid-Holocene (2,000–3,000 years ago) under slightly (c. 0.5 m) higher sea levels than present (Kench et al., 2020). Therefore, their development may have been characterised by more intense and frequent flooding than at present (East et al., 2018) and the current flooding regime may not reflect the contemporary sea-level position in relation to the island elevation.

Flooding occurred during a relatively modest spring high tide (c. 0.4 m MSL), but the high tide was augmented by a residual water level of c. 0.15 m, and the combined result was one of the highest water levels of 2022 (0.53 m MSL). The high water level coincided with the most energetic wave conditions experienced in 2022, characterised by an MFWAM-modelled offshore wave height $H_{s,o}$ of 3.3 m, which has a 10-year return period. A key contributing factor to the flooding was the long wave period during the peak of the storm of $T_p = 20.5$ s, confirming the notion that flooding in the Maldives is primarily associated with long swell events (Wadey et al., 2017).

Hydrodynamic data collected over a 2-month period immediately following the event on one of the affected islands, Fioyaree, were used to calibrate and validate the phase-resolving 1D XBeach model (XB-NH). A high roughness for the distinctive spur-and-groove section, roughly located between 0 m MSL and 10 m water depth, was required to bring the model results in line with the observations of waves and water level across the reef platform.

A Colebrook-White roughness $cf$ of 0.15 was allocated to the spur-and-groove section, and a $cf$ value of 0.01 for the deeper subtidal region and the reef platform. The model results are sensitive to the $cf$ value of the spur-and-groove section, with increased or decreased roughness having an opposing effect on wave setup and IG and VLF wave heights, as also shown by Van Dongeren et al. (2013). The calibrated value of $cf$ found in this study is, however, in line with previous modelling studies representing high coral cover (e.g., Storlazzi et al., 2019; Roelvink et al., 2021) and lies well within the range of previously reported $cf$ values of 0.002 (Quataert et al., 2020) to 0.4 (Klaver et al., 2019).

Model calibration and validation were conducted, but there were four main shortcomings to this part of the study. (1) The wave height derived from wave data collected in 12 m water depth (and linearly deshoaled to 25 m) was 63% of the MFWAM-modelled wave height from the deep water node (and shoaled to 25 m water depth). The large difference between offshore and nearshore wave height elicits caution when applying modelled offshore wave conditions to model nearshore hydrodynamics in atoll island settings in the absence of observational wave data. (2) This study uses a 1D hydrodynamic model, and it would have been more appropriate to set up a two-dimensional (2D) model on the island-scale. According to Van Dongeren et al. (2013), the use of a 1D XB-NH model requires a high value of $cf$ to reduce overestimation of wave setup at the expense of underestimation of IG and VLF wave height predictions. This was indeed observed, as the calibrated model overestimated the wave setup $\eta$ (positive bias of c. 0.1 m) and underestimated the combined IG and VLF wave height $H_{s,IGVLF}$ (negative bias of c. 0.1 m). It should be pointed out, however, that the 2D XB-NH model calibrated by Quataert et al. (2020) also shows a significant under-prediction of the VLF wave height (see their Figure 6). (3) The field data did not include run-up information, but the model results suggested that run-up

height can be parameterised by the TWH, defined as the sum of wave setup and SS, IG and VLF wave height measured at the toe of the island beach: $\eta + H_{s,SS} + H_{s,IG} + H_{s,VLF}$. Therefore, the model was optimised to minimise the bias in predicting TWH. Nevertheless, the modelled run-up elevation during the 1 July event was lower than the elevation of a gravel ridge present along most of the ocean-side of the island, suggesting that run-up may have been under-predicted. Field observations (wrack lines, overwash deposits and flood marks on buildings), on the other hand, indicated that island flooding mainly occurred along the island margins, where the gravel ridge is lower, and through localised depressions in the gravel ridge resulting from pedestrian access and following breaching of the gravel ridge. It is inferred on the basis of the combined evidence that the 2% exceedance run-up elevation $R_{2\%}$ during the peak of the flooding event may have been underestimated by up to 0.3 m, most likely due to an under-estimation of the combined IG-VLF wave motion. (4) Consistent with all previous local (Amores et al., 2022), regional (Taherkhani et al., 2020) and global (Vousdoukas et al., 2018) predictions of future coastal flooding, the present study also ignores any natural adjustments to the island morphology due to flooding. Sediment deposition due to overwash processes can result in an increase in island elevation (Tuck et al., 2019a; Tuck et al., 2019b; Masselink et al., 2020; Tuck et al., 2021; Roelvink et al., 2025) and may represent a natural adaptation to increased flood risk.

The calibrated and validated XBeach model was used to develop a 2% exceedance run-up elevation predictor that could potentially be used for evaluating future flood risk. Comparison between wave height and water level characteristics associated with major historic storm events and the run-up nomogram (Figure 15) explains why, in the past, flooding on Fioyaree was rare and will become increasingly common in the future. Specifically, in 1990, MSL was 0.15 m lower than currently, and a wave run-up *elevation* of 1.75 m MSL (run-up *height* of 1.25 m) would have been required to result in island flooding. None of the historic events exceeded this value; therefore, coastal flooding was extremely rare. In 2022, for which MSL was set to 0 m, a wave run-up *elevation* of 1.6 m MSL (*run-up* height of 1.1 m) led to island flooding. Of all the events since 1990, only the 1 July flooding event (and perhaps one additional event in May 2004) would have exceeded this value, making coastal flooding a c. 1:25-year event. In 2050, MSL is expected to be 0.2 m higher than in 2022, according to SSP-4.5 (IPCC, 2021), and a wave run-up *elevation* of 1.4 m MSL (run-up *height* of 0.9 m) would result in island flooding. Ten of the historic events over the 1990–2024 period would exceed this value, thereby, causing coastal flooding every few years. The expected increase in frequency of coastal flooding in the Maldives over the next few decades requires atoll and island authorities in the Maldives to act swiftly to adapt to future flood risk.

**Open peer review.** To view the open peer review materials for this article, please visit http://doi.org/10.1017/cft.2025.10013.

**Supplementary material.** The supplementary material for this article can be found at http://doi.org/10.1017/cft.2025.10013.

**Data availability statement.** Raw field data and numerical model parameters and results are publicly available and can be accessed via 10.24382/ce07577f-c55d-44aa-bdb7-0e06aeb72a5c (Masselink, 2024).

**Acknowledgements.** The authors would like to thank Liane Brodie, Kit Stokes, Daniel Conley and Peter Ganderton for helping out in the field and Lauren Bierman for identifying the source of the large tidal residual during the 1 July event. We thank LAMER for their partnership, support and use of resources during the field campaigns, especially Kandahalagalaa Ali Zuhair for his local insight, logistical support, skippering and overall assistance in the collection of field data. The authors also thank the Fioyaree island community for allowing them to make measurements and install instruments on their island.

**Author contribution.** GM conceived of the idea, received funding, collected field data, conducted the data analysis and carried out the numerical modelling. TP and TS collected field data, supported the data analysis and assisted with the writing. FR and RMC supported the numerical modelling and assisted with the writing.

**Financial support.** This research was supported by EPSRC Research Grant: Natural Adaptation of Atoll Islands to Sea-Level Rise Offering Opportunities for Ongoing Human Occupation (EP/X029506/1) awarded to Professor Gerhard Masselink, University of Plymouth.

**Competing interests.** The authors declare none.

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
