## [Reviewer Report]

Review for Cambridge Prisms: Coastal Futures

Manuscript ID: CFT-2025-0014

Title: Numerical modelling of the 1 July 2022 flooding event, Southwest Huvadhoo Atoll, Maldives: Implications for the Future

Article Type: Research Article

Overall

This paper aims to present and model a recent wave-driven flooding event in the Maldives. The model is validated using field observations and results confirm that extreme flooding events will likely become much more frequent in future. The topic is highly relevant and timely with growing concerns about future habitability of atoll island nations, and fits well within the scope of the journal. The paper is very well written, and is considered an important contribution to the field. A number of minor improvements or considerations are listed below.

Use of tide gauge data

The tide gauge record from Gan (Addu Atoll) was used to characterise tide and surge levels. However, this gauge 100 km away and while it is understood there is no other data available, the authors should provide more motivation and discussion on the representativeness of this data for this specific study.

Methods description

The Methods section of the paper is extremely brief and reference is made to two separate documents (support information) that provide information about the field data collection (S1) and numerical modelling (S2). To a minimum it would be helpful to the reader to have the main information listed in the main text, with only details left for the supporting information. In particular, information on model setup and choices including calibration coefficients would ideally be mentioned here and not only in the supporting information.

Methods – morphodynamic modelling approach

While detailed description is provided in S2, some aspects on the modelling are currently insufficiently explained or motivated. For instance, it is unclear why the authors use sediment properties that vary two orders of magnitude while sediment size data was available / collected and there are no simulations with sediment data from the field as input. Also, since the model calibration/validation does not include a comparison with morphological data, it would be suggested to include a sensitivity analysis on the key coefficients related to sediment transport, such as the phase angle.

Minor edits

Line 49: ‘2025’ should be ‘2050’

Line 50: this line suggests the flooding is primarily due to waves but following the paper the alignment with spring high tide was essential. ‘… result of a distant-swell event in combination with spring high tide.’

Line 101: tsunamis

Line 169: perhaps clarify that Gan is on Addu Atoll

Line 204/205: was this also found in the Fares-Maathoda tide data?

Figure 4: add months to horizontal label

Line 308-310: is the sediment size based on literature, if collected as part of this study, why not included in Supporting information S1?

Figure 9: confusing, modelled Hs and Tp but is the offshore boundary condition? Panel a description says only measured water level but based on main text only blue line is measured (model boundary condition), all other 3 lines are model predcitions?

Section 4.3: why not just run a simulation with the actual sediment properties? Why run a fine-sand and a gravel case only (both D50 and K two orders of magnitude different)?

Sensitivity to phase angle?

---

## [Reviewer Report]

I struggled with this paper for two reasons: (1) it was not clear how the outcomes achieved the stated objectives; (2) it was not clear what the aspects were generalisable to other locations. I think some of this could be fixed by finding an aim that better reflects what is possible from the available data and methods. The mapping of the impact of the events was very useful because it there do not seem to many studies that do this following events. The introduction of the paper indicates that the model was well calibrated and verified, but no evidence of this is provided in the results. There is an extensive supplement, but that seemed to indicate that the model did not calibrate well at all (Figure S2.3)? There were no run-up observations, and so the model could not actually be calibrated or verified in this respect. Even so, believing in the quality of the model is central to the paper, and so showing this clearly and transparently is very important to building to the final conclusions. The discussion starts to address the reasons for poor model fit (so contracting the last paragraph of the introduction).

There likely will be a well constructed and clearly argued paper here eventually, but it is very hard to follow now. It needs quite a major revision to accomplish this.

There were also quite a few things that were confusing. For example, I don’t quite understand Figure 11. I understand what is written in the text, but not the caption. Do you mean to say that you modelled all 158 wave events with Xbeach? In the text you say “parameter space based on their offshore peak wave height Hs,o” but in the caption you say (“peak modelled Hso”). So the colorscale is run-up modelled using the offshore peak wave height as forcing, where as the circles are the modelled wave heights (modelled using Xbeach). A bit more clarity in the wording would help a lot.

I also don’t quite understand what MSL means in the caption of Figure 11. Do you mean the R%>1.5m [relative] to MSL. Or do you mean that you changed the MSL to reflect different sea level rise projections in your simulations?

Some more minor comments:

Abstract: “predicted reasonably well.” Please provide a more objective statement here. What is reasonable? What accuracy and how was that assessed? (also in the impact statement)

L72: Odd wording…what does it mean to say that ‘wave motion is released as runup’ ? Wave motion along the coastline is expressed as a runup signal.

In general, the text has a lot of ‘lonely pronouns’, eg it is released,….this is an oversimplification….it is widely considered.

There are also a few colloquial expressions, such as previous studies ‘point towards’ , ‘linearly deshoaled’

L163: the topography (c. 1:12) drops [steeply] down

L190: ‘Measured water level and wave conditions are not available for the study site.’

L202-204: It would be good to include more standard exceedance measures like annual exceedance probability or peaks over threshold. Did you consider using the skew surge instead, because some of the residual can generally be attributed to poorly resolved tidal constituents, particularly during higher water levels which cause nonlinear feedbacks into the tidal height.

L223-233: 158 storms, what threshold did you use to remove clustered storm events? E.g. ones that were greater than 3-days apart? I don’t think there is a standard time window, but people usually use something.

L248: Was the only verification of the storm output from visual observations? Sometime spray such things can make the visual signature (e.g. wrack lines) quite a bit higher than the runup extreme. Or where only the sedimentary overwash deposits mapped? I see later in the methods, that an in-situ deployment were used to calibrate the model. I think it would be good to make this clear in the last paragraph of the introduction “(2) to reproduce the flooding event using a phase-resolving numerical model (XBeach-NH) that is calibrated and validated with local hydrodynamic data;” Sounds like it was validated during storm conditions.

Figure 10 and morphodynamic modelling. How do you know this is right? It must be highly sensitive to the bed composition and interlocking of the coral pieces. How did you decide what was an appropriate setting in the model for this? This is discussed later, but to believe this result, the model needs to be properly presented.

447-449: Again, if the roughness is a discussion point, the calibration details should be presented in the results and not in the supplement. Also the discussion on bias is based on material not presented in the results.

512: I hope they don’t create an early warning system using unverified predictors!

Figure 1 has no source attributed, also Figure 8.

Figure 3: Please add the maximum annual water level to panel A.

---

## [Reviewer Report]

The authors have handled the comments well and the new manuscript is a considerable improvement. I have no further objections regarding publication in Coastal Futures.

---

## [Reviewer Report]

The paper is much more readable and focused now, with the removal of the morphodynamic modelling, and moving much of the modelling from the supplement to the main paper. Given the lack of information about storm events from atoll environments, and the extensive deployment of sensors in this remote location, I believe that there will be interest in this paper.

There are only a few minor things that would improve the paper still. Figures around calibration/validation of the model are presented in a confusing way. Usually a model-fit scatter plot has the same variable on each access, one model, one data. By the nature of the scatter, this will show the bias, RMSE and r values, usually the 1:1 and the line of best fit is shown, with the prediction provided by the optimum settings, rather than also showing the results from non optimal settings. Instead, 10b shows modelled runup against TWH (which is useful because it shows the source of error in not being able to observe the runup). Figure 11 shows model fit, but never plots the predicted against the observed, making it difficult to see specifically where the model fits well or poorly? At least Fig 11 should show predicted against observed. In fact, figures 10&11 could be combined, with the calibration and verification shown on the same figures with different coloured symbols. There could be different panels for (1)TWL, (2)Setup, (3)HsSS (4)HsIF and (5) HSVLF, and the 6th plot could be 11b. The model fit table in the supplement should also include the Brier skill score.

Otherwise, I have no further comments, and look forward to seeing the paper published.

---

## [Editor Report]

The authors have made substantial revisions to the manuscript. I believe that this can be accepted once the recommended minor revisions are complete, without the need for an additional peer review round.

---

## [Editor Report]

I have already added my comments in the Handling Editor section. The authors have addressed the few outstanding issues with the paper, and I am happy to accept this without the need for further reviews.